# AGC-Drive: A Large-Scale Dataset for Real-World Aerial-Ground Collaboration in Driving Scenarios

**Yunhao Hou[1], Bochao Zou[1,*], Min Zhang[2], Ran Chen[1], Shangdong Yang[2], Yanmei Zhang[2]**
**Junbao Zhuo[1], Siheng Chen[3], Jiansheng Chen[1], Huimin Ma[1,*]**
[1]University of Science and Technology Beijing
[2]Xiamen NEVC Advanced Electric Powertrain Technology Innovation Center
[3]Shanghai Jiao Tong University
*Corresponding author: {zoubochao, mhmpub}@ustb.edu.cn

## Abstract

By sharing information across multiple agents, collaborative perception helps autonomous vehicles mitigate occlusions and improve overall perception accuracy. While most previous work focus on vehicle-to-vehicle and vehicle-to-infrastructure collaboration, with limited attention to aerial perspectives provided by UAVs, which uniquely offer dynamic, top-down views to alleviate occlusions and monitor large-scale interactive environments. A major reason for this is the lack of high-quality datasets for aerial-ground collaborative scenarios. To bridge this gap, we present AGC-Drive, the first large-scale real-world dataset for Aerial-Ground Cooperative 3D perception. The data collection platform consists of two vehicles, each equipped with five cameras and one LiDAR sensor, and one UAV carrying a forward-facing camera and a LiDAR sensor, enabling comprehensive multi-view and multi-agent perception. Consisting of approximately 80K LiDAR frames and 360K images, the dataset covers 14 diverse real-world driving scenarios, including urban roundabouts, highway tunnels, and on/off ramps. Notably, 17% of the data comprises dynamic interaction events, including vehicle cut-ins, cut-outs, and frequent lane changes. AGC-Drive contains 350 scenes, each with approximately 100 frames and fully annotated 3D bounding boxes covering 13 object categories. We provide benchmarks for two 3D perception tasks: vehicle-to-vehicle collaborative perception and vehicle-to-UAV collaborative perception. Additionally, we release an open-source toolkit, including spatiotemporal alignment verification tools, multi-agent visualization systems, and collaborative annotation utilities. The dataset and code are available at https://github.com/PercepX/AGC-Drive.

## 1 Introduction

Perception serves as a critical foundation for decision-making and safety in autonomous driving and multi-agent systems, especially in dynamic scenes with occlusions, long-range detection, and rapid response needs. To enhance perception completeness, existing cooperative perception systems mainly focus on Vehicle-to-Vehicle (V2V) [1–3][4, 5] and Vehicle-to-Infrastructure (V2I) [3–5][6–8][9–11] frameworks. V2V cooperative perception mitigates local occlusion issues by enabling information exchange among nearby vehicles. However, as all sensors remain at ground level, V2V systems struggle with dense traffic, occlusions, complex intersections, and limited perception range [6, 12]. Their performance is highly dependent on vehicle distribution and communication reliability. V2I cooperative perception utilizes roadside units (RSUs) to enhance sensing capabilities at critical points such as intersections and road segments [6, 8, 9]. Nevertheless, V2I systems face inherent

39th Conference on Neural Information Processing Systems (NeurIPS 2025) Track on Datasets and Benchmarks.

limitations in deployment cost, fixed coverage, and adaptability to dynamic environments, making them unsuitable for large-scale open roads or rapidly evolving traffic scenarios.

Unlike V2V and V2I systems, Aerial-Ground Cooperative (AGC) Perception introduces overhead Unmanned Aerial Vehicle (UAV)-based sensing, offering dynamic, adaptive, and high-altitude global perspectives to complement ground-based systems. UAVs can flexibly cover target areas, dynamically alleviate perception blind spots, enhance long-range target observation, and improve multi-object occlusion reasoning in complex traffic environments. This makes AGC perception a valuable complement to existing V2X systems, particularly in scenarios involving open roads, dynamic intersections, dense traffic, and emergency scenarios.

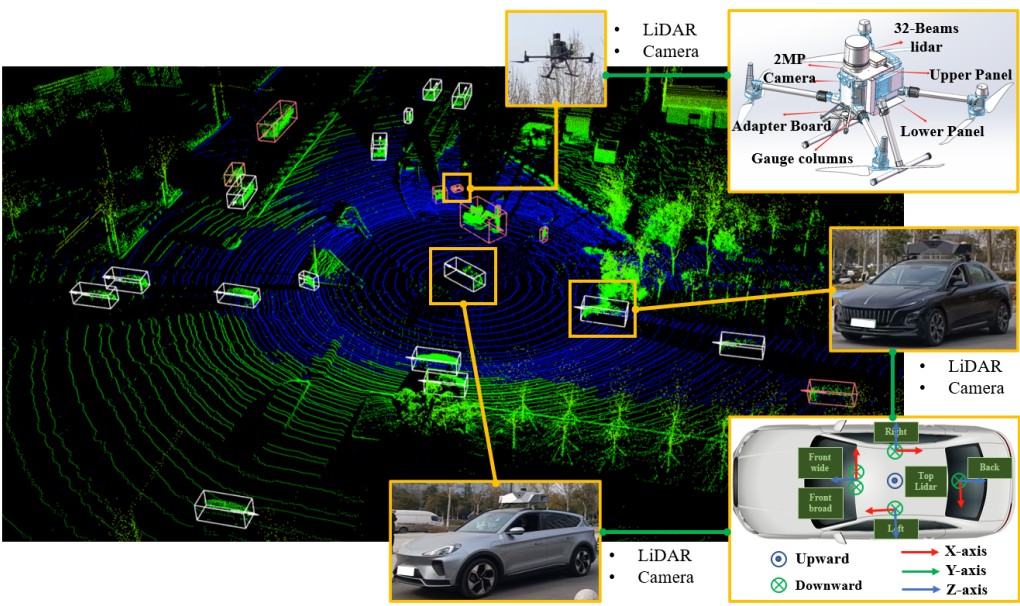

Figure 1: Collaborative data collection with two vehicles and a UAV. Each vehicle is equipped with one LiDAR and five cameras. The UAV carries a LiDAR and a camera system. The top-right inset shows the custom UAV sensor setup, and the bottom-right inset illustrates the vehicle's sensor layout.

Although several datasets [13–16, 12] have introduced UAVs into collaborative perception, they either focus on UAV-to-UAV cooperation [13, 14] or are collected in simulated environments [13–15, 12]. The only available real-world dataset [16] provides 2D annotations, but it is not collected in driving scenarios. Due to hardware limitations, none of these existing datasets include LiDAR-equipped UAVs. To bridge this gap, we introduce AGC-Drive, a dataset collected by a system consisting of two vehicles and one UAV (Figure 1) over approximately three months. From over 80 hours of collected data, we selected 350 sequences covering 14 scene categories. Each sequence contains 100 sets of data sampled at 10 Hz, with each set including 14 LiDAR frames and image frames, resulting in over 720K annotated 3D bounding boxes. Notably, we carefully designed the data collection routes to ensure a wide range of road conditions and vehicle interaction scenarios, covering high-risk environments such as urban roundabouts, highway tunnels, on/off ramps, and rural construction zones. Approximately 17% of the data involves dynamic events like vehicle cut-ins, cut-outs, and dense lane changes.

To support broad research in collaborative perception, we organized our dataset into two dedicated sub-collections: AGC-V2V for vehicle-to-vehicle collaboration, and AGC-VUC for vehicle-to-UAV collaboration. Additionally, to address the lack of UAV-based 3D object detection datasets using airborne LiDAR, we plan to introduce AGC-U3D, a carefully curated subset for UAV 3D object detection tasks.

Our main contributions are summarized as follows:

- We present AGC-Drive, the first real-world vehicle-vehicle-UAV collaborative perception dataset for driving scenarios, featuring time-synchronized multi-agent data collection and fused 360° ground and aerial views. It includes two sub-datasets: AGC-V2V and AGC-VUC.

- We provide over 720K annotated 3D bounding boxes for 13 categories, covering 80K LiDAR frames and 360K multi-view images, collected across 14 types of road environments and dynamic interaction scenarios such as lane changes and overtaking.

- We report benchmarks for two 3D perception tasks and release an open-source toolkit for spatiotemporal alignment, multi-agent visualization, and collaborative annotation.

## 2 Related work

With the rapid development of collaborative perception, an increasing number of high-quality datasets have been released. OPV2V [1] was the first collaborative perception dataset, featuring vehicle-to-vehicle (V2V) synthetic data. V2X-Sim [3] and V2XSet [4] extended this to vehicle-to-infrastructure (V2I) and V2V scenarios, but remained in simulated environments. DAIR-V2X [6] introduced the first large-scale, multi-modal, multi-view real-world V2I collaborative perception dataset. V2X-Seq [7] further contributed the first large-scale sequential collaborative dataset. Subsequently, five large-scale real-world datasets — V2V4Real [2], RCooper [8], TUMTraf-V2X [9], HoloVIC [10], and V2X-Real [5] have advanced research in vehicle-centric collaborative perception. Most recently, V2X-R [11] became the first dataset to incorporate 4D radar into collaborative perception.

Meanwhile, recent releases such as CoPerception-UAV [13] and UAV3D [14] reflect growing interest in incorporating UAVs into collaborative perception. However, both focus solely on UAV-to-UAV collaboration in simulated environments. More recently, V2U-COO [17] and Griffin [12] proposed aerial-ground collaborative datasets, but they are also synthetic and lack UAV LiDAR data due to hardware limitations. Although CoPeD [16] is a real-world aerial-ground collaborative dataset, it only provides monocular camera data with simple 2D annotations automatically generated by baseline models.

In contrast, AGC-Drive is a large-scale real-world dataset designed to support collaborative perception between aerial UAVs and ground vehicles. The UAV platforms are custom-designed and carefully equipped with LiDAR sensors. Table 1 summarizes the comparison with related datasets. It is worth noting that, similar to nuScenes [18], We annotate occlusion levels for each object, a detail often neglected in prior collaborative perception datasets. Moreover, our dataset covers 14 diverse scene categories under varying lighting conditions, with a day-to-night ratio of 8:2, significantly enhancing data diversity and robustness for perception tasks.

Table 1: Comparison of representative single-UAV and cooperative perception datasets. V/Veh = Vehicle, I/Inf = Infrastructure, U = UAV. "V2V&I" = V2V+V2I, "V2V&U" = V2V+V2U. C, L, R = Camera, Lidar, Radar. "MvCamera" = Multi-view Camera, "UAV-L" = UAV with LiDAR.

| Mode | Dataset | Year | Source | Agent | Sensor | scenario types | 3D boxes | Classes | MvCams | Driving | UAV-L |
|------|---------|------|--------|-------|--------|---------|---------|---------|--------|---------|-------|
| V2V | OPV2V [1] | 2022 | Sim | Veh | C & L | 6 | 230K | 1 | ✓ | ✓ | × |
|  | V2V4Real [2] | 2023 | Real | Veh | C & L | - | 240K | 5 | ✓ | ✓ | × |
| V2I | DAIR-V2X [6] | 2022 | Real | Veh & Inf | C & L | - | 464K | 10 | × | ✓ | × |
|  | V2X-Seq [7] | 2023 | Real | Veh & Inf | C & L | - | - | 9 | × | ✓ | × |
|  | Rcooper [8] | 2024 | Real | Veh & Inf | C & L | - | - | 10 | × | ✓ | × |
|  | TUMTraf-V2X [9] | 2024 | Real | Veh & Inf | C & L | - | 29.3K | 8 | × | ✓ | × |
|  | HoloVIC [10] | 2024 | Real | Veh & Inf | C & L | - | 11.4M | 3 | × | ✓ | × |
|  | V2X-R [11] | 2025 | Real | Veh & Inf | C & L & R | - | - | 5 | × | ✓ | × |
| V2V&I | V2X-Sim [3] | 2022 | Sim | Veh & Inf | C & L | - | 26.6K | 1 | ✓ | ✓ | × |
|  | V2XSet [4] | 2022 | Sim | Veh & Inf | C & L | 5 | 230K | 1 | ✓ | ✓ | × |
|  | V2X-Real [5] | 2024 | Real | Veh & Inf | C & L | - | 1.2M | 10 | ✓ | ✓ | × |
| UAV | VisDrone [19] | 2018 | Real | UAV | C | - | 10.2K | 10 | × | × | × |
|  | UAVDT [20] | 2018 | Real | UAV | C | - | 841.5K | 3 | ✓ | ✓ | × |
| U2U | CoPerception-UAV [13] | 2023 | Sim | UAV | C | - | 1.6M | 21 | ✓ | ✓ | × |
|  | UAV3D [14] | 2023 | Sim | UAV | C | - | 3.3M | 17 | ✓ | ✓ | × |
| V2U | V2U-COO [17] | 2024 | Sim | Veh & UAV | C | - | - | 4 | × | ✓ | × |
|  | CoPeD [16] | 2024 | Real | Veh & UAV | C & L | 2 | × | 1 | × | × | × |
|  | Griffin [12] | 2025 | Sim | Veh & UAV | C & L | 4 | - | 3 | ✓ | ✓ | × |
| V2V&U | **AGC-Drive(Ours)** | **2025** | **Real** | **Veh & UAV** | **C & L & R** | **14** | **720K** | **13** | ✓ | ✓ | ✓ |

# 3 The AGC-Drive Dataset

To bridge collaborative vehicle-to-vehicle and aerial-ground perception research and to establish a comprehensive 3D traffic perception framework, we present AGC-Drive — a large-scale, multimodal, multi-view, and multi-scenario dataset featuring well-annotated 3D bounding boxes for innovative research on UAV-vehicle collaboration. In this section, we describe the data acquisition devices, coordinate system design, multi-sensor calibration, scene selection strategies, detailed data collection, annotation processes and pose refinement, as well as privacy protection considerations.

## 3.1 Setup

**Sensors.** The data acquisition system consists of two instrumented vehicles and one UAV: a) Vehicle-mounted sensors. Each vehicle is equipped with a 128-beam LiDAR and five high-resolution cameras. Notably, the front-facing cameras are configured with two different focal lengths to capture both detailed road surface information and broader traffic scene context; b) UAV-mounted sensors. The UAV platform is based on a modified DJI M350 RTK, equipped with a 32-beam LiDAR and a high-resolution downward-facing camera. The sensor configuration is illustrated in Fig.1, with detailed specifications summarized in Table 2.

Table 2: Key Sensor Specifications in AGC-Drive.

| Agent | Sensor | Sensor Model | Detail |
|---|---|---|---|
| 2*Vehicle | LiDAR | RoboSense Ruby Plus(*1) | 128 beams, 10Hz capture frequency, 360°horizontal FOV, -25°to +15°vertical FOV, < 200m range |
| | Camera | Sensing Cmaera(*5) | front-wide: SG8S-AR0820C-5300-G2A-Hxxx, 8MP, HFOV30°, front-broad: SG8S-AR0820C-5300-G2A-Hxxx, 8MP, HFOV120°, left&right: SG2-AR0231C-0202-GMSL-Hxxx, 2MP, HFOV100°, back: SG2-AR0233C-5200-G2A-Hxxx, 2mp, HFOV121° |
| | GPS&IMU | Intelligent Car Built-in GPS System(*1) | 100HZ |
| UAV | LiDAR | RoboSense Helios32(*1) | 32 beams, 10Hz capture frequency, 360°horizontal FOV, -55°to +15°vertical FOV,< 150m range |
| | Camera | USB Camera(*1) | front: RER-USBGS1200P02, 2MP, HFOV120° |
| | GPS&IMU | DJI M350 RTK Built-in GPS System(*1) | GPS + GLONASS + BeiDou + Galileo, 100HZ |

**Coordinate System.** The AGC-Drive dataset defines four types of coordinate systems: LiDAR coordinate system, camera coordinate system, image coordinate system, and world coordinate system. Each agent is equipped with its own LiDAR and camera coordinate systems. The camera coordinate system is aligned to the corresponding LiDAR frame via a camera-to-LiDAR calibration process. Then, the pose information from each agent's GPS/IMU is used as an initial transformation, and all LiDAR coordinate systems are registered to a unified world coordinate system through point cloud registration. The calibration results are illustrated in Fig. 1.

**Calibration.** To ensure accurate spatial alignment between the LiDAR point cloud and multiple cameras, we perform extrinsic calibration for each camera with respect to the LiDAR sensor. First, the intrinsic parameters of each camera are obtained using a standard checkerboard calibration. Then, we collect synchronized images and LiDAR data of a calibration target visible in both modalities. 3D-2D correspondences are established by detecting feature points in the images and extracting the corresponding points from the LiDAR point cloud. Finally, the extrinsic transformation matrices are estimated using a Perspective-n-Point (PnP) algorithm [21], followed by visual verification through point cloud projection onto the image plane.

**Scenario Planning.** To ensure the representativeness and utility of our dataset, we follow the Task-driven Scenario Taxonomy [22] and Operational Design Domain (ODD) definitions [23]. We also reference popular autonomous driving datasets such as Waymo Open Dataset [24] and nuScenes [18]. Scenarios are designed based on common driving tasks including lane keeping, lane changing, car-following, intersection crossing, roundabout navigation, construction detours, and ramp merging. Combined with ODD, they are categorized into urban, highway, and rural environments, further covering typical cases like straight roads, curves, intersections, construction zones, tunnels, and frequent lane changes. In total, 14 representative driving scenarios are constructed (Fig. 2), providing comprehensive coverage for both routine and high-risk conditions.

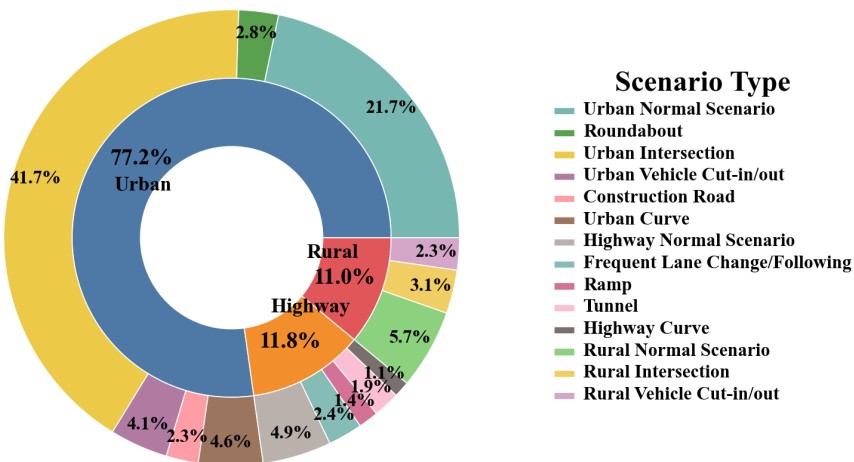

Figure 2: Distribution of Driving Environment and Scenario Types.

## 3.2 Data Acquisition

**Collection.** We deployed two equipped vehicles and a UAV operated by professional pilots to collect synchronized data. Each agent locally records data frames, with timestamps aligned to its GPS time, achieving unified time sources across all agents. Data were collected along predefined routes over 2 months under varying day and night conditions, accumulating over 80 hours of driving logs. From the raw data, we curated 350 representative segments (10s each) across 14 scenarios, resulting in 80K LiDAR frames and 360K camera images with annotations. Keyframes were extracted at 10 Hz to construct AGC-V2V, while segments involving UAV participation were used to create AGC-VUC for vehicle-UAV collaborative perception tasks. Additionally, we plan to extracted UAV LiDAR data to support UAV-based 3D perception research.

**Annotation.** We customized an annotation platform based on the open-source tool Xtreme1 [25] to meet the specific needs of our dataset. A team of 100 annotators was recruited, with multiple rounds of expert review to ensure annotation quality. On average, each frame underwent two rounds of review and revision. Our dataset provides 3D annotations for 13 object categories, including pedestrian, rider, motorcycle, bicycle, tricycle, car, truck, van, bus, road obstacles, traffic cones, and traffic signs. Each object is labeled with a 9-DoF 3D bounding box, consisting of (x, y, z) for the box center, (l, w, h) for size, and roll, pitch, yaw for orientation. Furthermore, each 3D bounding box is projected onto the corresponding camera images to generate two types of 2D annotations: an 8-vertex 2D polygon representing the projected 3D box corners, and a 4-vertex 2D rectangle corresponding to the minimum enclosing rectangle of the projected box. Additionally, each object is assigned one of three occlusion levels: visible (0–20%), partially occluded (20–50%), or heavily occluded (over 50%).

**Relative Pose.** In this work, we estimate the relative poses between three agents: two vehicles and a UAV, using their respective GPS, IMU, and LiDAR data. Initially, GPS and IMU data provide the initial pose estimates for each agent. These initial poses serve as the starting input for the Iterative Closest Point (ICP) algorithm [26]. The ICP algorithm is then used to perform pairwise point cloud registration, aligning the LiDAR point clouds of the vehicles and UAV. This process calculates the relative poses between each pair of agents based on the aligned point clouds. After ICP registration, the relative poses are refined and corrected to improve accuracy. Finally, the corrected poses are transformed into the ego vehicle's coordinate frame, ensuring consistent spatial alignment across all agents. This method enables robust and accurate relative pose estimation, which is crucial for multi-agent cooperative perception tasks.

**Privacy Protection.** Prior to public release, all sensitive information in the dataset was anonymized. We removed all location metadata, including road names, map data, and GPS information, to comply with legal and ethical regulations. In addition, a professional annotation tool was used to blur potential privacy-related content, such as traffic signs, license plates, and human faces, to ensure protection of personal privacy.

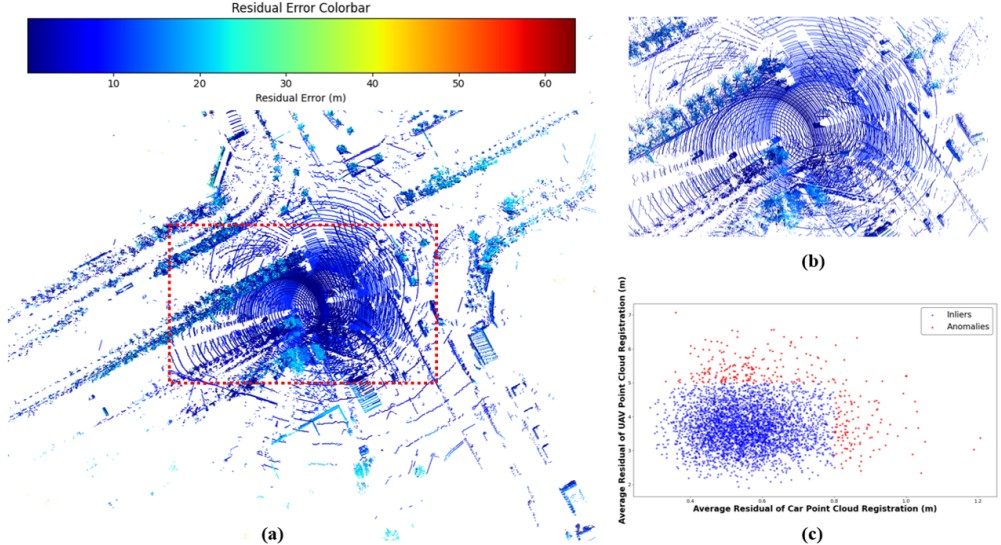

Figure 3: (a) Residual heatmap of registration results (blue: high accuracy, red: low accuracy). (b) Zoomed view of the dense object region in (a). (c) Scatter plot of average residuals for 4000 randomly sampled points after registering both drone and vehicle point clouds to the ego vehicle.

# 4 Task and Benchmark

We benchmark AGC-Drive on collaborative 3D object detection tasks, including vehicle-to-vehicle cooperation, vehicle-UAV cooperative(VUC) 3D object detection.

**Dataset Format and Split.** Our AGC-Drive dataset follows the OPV2V [1] format, containing 14 road types with multiple scenes per type (distribution shown in Fig. 2). Each scene consists of 100 groups of synchronized multi-agent data, collected by 1 drone and 2 vehicles equipped with a total of 14 sensors. In total, we collect 350 scenes, with 320 for training, 30 for validation and testing.

**Input and Ground Truth.** For each agent, the input includes synchronized LiDAR point clouds captured at 10 Hz. The shared data from neighboring vehicles are fused in the bird's eye view (BEV) space for cooperative perception. The ground truth consists of 9-DoF 3D bounding boxes annotated for 13 categories (car, truck, pedestrian, cyclist, etc.) with additional occlusion status labels (visible, partially occluded, heavily occluded) following the nuScenes convention [18].

**Benchmark Frameworks.** We benchmark six representative cooperative perception frameworks, all of which adopt PointPillars [27] as the detection backbone for a fair comparison:

- **Late Fusion**: Directly shares raw point cloud data among agents before feature extraction.
- **Early Fusion**: Independently detects objects and shares detection results among agents.
- **V2VNet** [1]: A multi-agent cooperative detection framework using intermediate feature fusion.
- **Cobevt** [28]: A cooperative BEV semantic segmentation framework based on sparse transformers, employing a feature aggregation module (FAX) to effectively fuse multi-view and multi-agent features.
- **Where2comm** [13]: A communication-efficient cooperative perception framework that guides agents to share only spatially sparse yet perception-critical information using a spatial confidence map, achieving a balance between perception performance and communication bandwidth.
- **V2X-ViT** [4]: A recent transformer-based cooperative perception framework leveraging BEV feature fusion with attention mechanisms.

**Experiments Details.** For all tasks, the models are adapted to support the AGC-V2V and AGC-VUC data format and sensor configurations. We adopt a unified BEV (Bird's Eye View) representation.

The perception range is set to $[-140.8\,\text{m}, 140.8\,\text{m}]$ along both the X and Y axes. All baseline models were trained with a batch size of 4 for 60 epochs, with a per-GPU batch size of 1 on a computing server equipped with 8 Nvidia L40 GPUs. We use the Adam optimizer with an initial learning rate of 0.001 and apply a cosine learning rate schedule. Each training run takes approximately 6 hours.

Following the evaluation protocols of nuScenes [18] and OPV2V [1], we report standard metrics including mAP@0.5 and mAP@0.7 for 3D object detection. Furthermore, to analyze the influence of UAV participation, we define the metric $\Delta_{\text{UAV}}$ as:

$$\Delta_{\text{UAV}} = \frac{1}{2}\left[\left(m_{0.5}^{V2U} - m_{0.5}^{V2V}\right) + \left(m_{0.7}^{V2U} - m_{0.7}^{V2V}\right)\right],$$

which represents the average performance improvement achieved by incorporating aerial perception.

## 4.1 Benchmark for V2V 3D object detection

**Problem Definition and AGC-V2V.** The goal of this task is to perform cooperative 3D object detection by leveraging information from multiple connected vehicles to enhance perception performance, especially in challenging scenarios such as long-range detection and occlusions. We construct the AGC-V2V benchmark by selecting collaborative scenes from the AGC-Drive dataset where two vehicles participate without UAV involvement. Each vehicle is equipped with a LiDAR sensor and five cameras. A total of 350 scenes, comprising 70K frames, are included for this benchmark.

**Quantitative Results.** Table 3 presents the 3D object detection results on AGC-V2V. As expected, early fusion achieves slightly better performance than late fusion, benefiting from access to fully aggregated features. Among intermediate fusion methods, Cobevt and V2X-ViT outperform others, indicating their stronger capability in feature aggregation. In contrast, V2VNet performs poorly at higher IoU thresholds, suggesting sensitivity to feature misalignment. Overall, performance remains moderate across all methods, likely due to challenges such as time delays and pose estimation errors in cooperative perception. The late fusion baseline exhibits the lowest accuracy, reflecting error accumulation during post-fusion processing.

Table 3: 3D Detection Performance (%) on AGC-V2V.

| Co-Mode | Model | mAP@0.5 | mAP@0.7 |
|---------|-------|---------|---------|
| Late | PointPillars[27] | 17.7 | 13.5 |
| Early | PointPillars[27] | 19.6 | 14.1 |
| Intermediate | V2VNet [1] | 18.4 | 5.7 |
| | Cobevt [28] | **46.1** | **41.7** |
| | Where2comm [13] | 39.3 | 31.5 |
| | V2X-ViT [4] | 44.1 | 36.6 |

**Qualitative results.** Figure 4 presents qualitative 3D object detection results of four intermediate collaborative baselines under several challenging scenarios, including highway tunnel occlusions, intersection occlusions with long-range targets, and complex road occlusions. We see that V2V cooperative perception effectively alleviates the challenges posed by long-range perception and occlusions. However, due to unaddressed pose errors and time delays, the localization accuracy of the predicted results is lower than that of single-vehicle detection, leading to a decline in overall performance metrics.

## 4.2 Benchmark for VUC 3D object detection

**Problem Definition and AGC-VUC.** This task aims to improve 3D object detection by leveraging the complementary perception capabilities of ground vehicles and UAVs. Unlike traditional V2V cooperation, the UAV provides a global top-down view that enhances occlusion handling and long-range detection. To support this, we construct the AGC-VUC benchmark by selecting cooperative scenarios in AGC-Drive that involve two vehicles and one UAV. Each scenario lasts approximately 10 seconds, sampled at 10Hz, resulting in 100 cooperative sequences.

**Quantitative Results.** Table 4 presents the 3D detection performance on AGC-VUC after incorporating the UAV into the cooperative system. A new column, denoted as $\Delta_{\text{UAV}}$, is added to quantify the

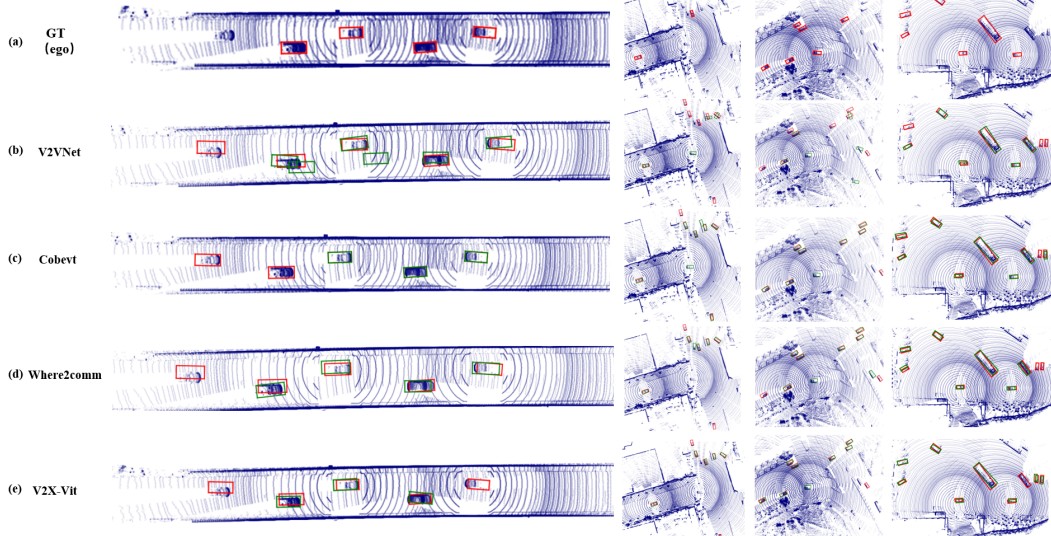

Figure 4: Point cloud visualization of V2V cooperative object detection results on the AGC-V2V dataset. Green bounding boxes denote predicted objects, and red bounding boxes indicate ground truth annotations.

impact of the UAV's participation on the perception performance. It can be observed that introducing the UAV improves the performance of all cooperative frameworks across all metrics. Notably, V2VNet achieves the largest improvement, with a $\Delta_{\text{UAV}}$ of +11.5. This is because V2VNet has the lowest baseline performance without the UAV, leaving more room for the UAV to enhance its perception capabilities.

Table 4: 3D Detection Performance (%) on AGC-VUC.

| Co-Mode | Model | V2V | | V2U | | |
| | | mAP@0.5 | mAP@0.7 | mAP@0.5 | mAP@0.7 | $\Delta_{\text{UAV}}$ |
|---|---|---|---|---|---|---|
| Intermediate | V2VNet [1] | 30.5 | 14.6 | **40.1** | **27.9** | **+11.5** |
| | Cobevt [28] | 42.3 | 36.9 | **42.9** | **37.5** | +0.6 |
| | Where2comm [13] | 42.6 | 30.7 | **44.2** | **32.0** | +1.5 |
| | V2X-ViT [4] | 38.3 | 28.7 | **42.6** | **33.9** | +4.8 |

**Qualitative results.** Figure 5 presents qualitative 3D object detection results of four intermediate collaborative baselines under several challenging scenarios. By comparing the visualization results with V2V baselines, we observe that integrating UAV perspective data effectively improves the ego vehicle's perception performance, particularly for distant and occluded objects in a larger area of the same scene.

## 5   Limitations

Since this is the first attempt to mount a vehicle-mounted LiDAR sensor on a UAV for real-time object detection data collection, we carefully considered factors such as the UAV's flight altitude, the LiDAR's weight, and the blind zone beneath the UAV caused by the LiDAR's vertical field of view during the system design. However, there were still some aspects that were insufficiently addressed. In our experimental validation, although the point cloud data collected from the UAV perspective contributed to a more comprehensive perception of large-scale scenes, its relatively sparse nature limited its ability to provide fine-grained perception assistance for collaborative tasks. We have recognized this issue and plan to explore potential upgrades or replacements for the UAV-mounted LiDAR before conducting large-scale dataset collection in the future. Our goal is for the UAV's point clouds not only to enhance large-scale scene awareness but also to offer more detailed, object-level perception information to support ground vehicles.

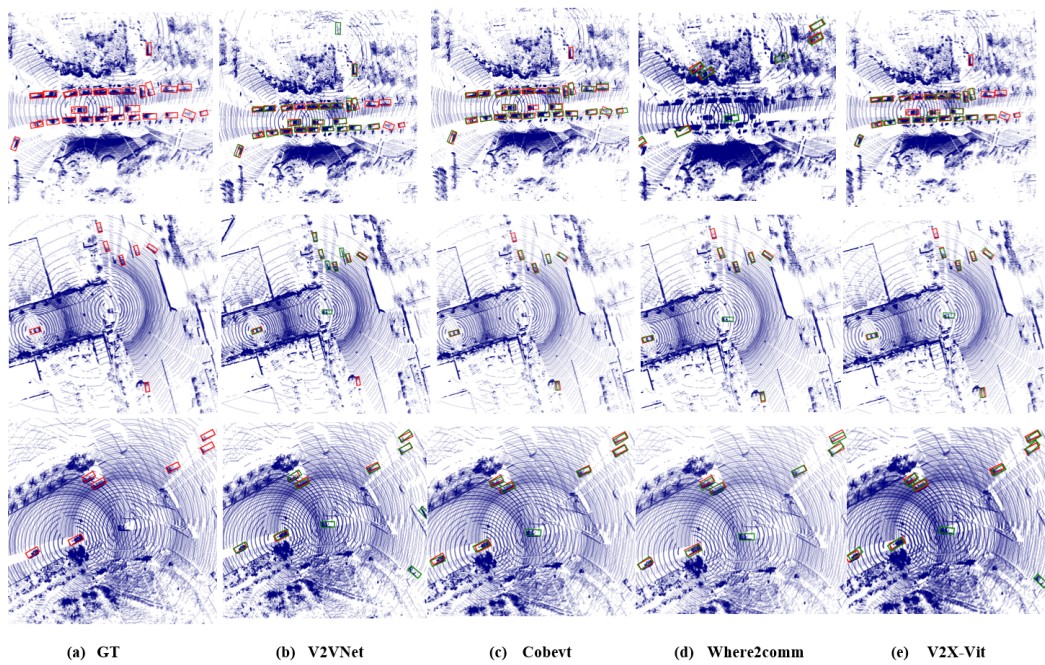

|  (a)  GT | (b)  V2VNet | (c)  Cobevt | (d)  Where2comm | (e)  V2X-Vit |

Figure 5: Point cloud visualization of VUC object detection results on the AGC-VUC dataset. Green bounding boxes denote predicted objects, and red bounding boxes indicate ground truth annotations.

# 6   Conclusion

We present AGC-Drive, a large-scale, multimodal dataset for collaborative perception between aerial UAVs and ground vehicles, collected in real-world environments. Compared to existing aerial-ground collaborative perception datasets, AGC-Drive is built upon real-world data with annotated 3D bounding boxes. Notably, it includes point cloud data collected from the UAV perspective, enabling collaborative perception at scene level. We further adapt several representative collaborative perception frameworks to our dataset and provide comprehensive benchmark results. It is worth noting that we retain data containing time delays and pose errors caused by point cloud registration, in order to better reflect the challenges encountered in real-world scenarios. All associated resources, including benchmark code, annotation tools, pose correction toolkits, and the complete AGC-Drive dataset, are publicly released. In addition, our raw data includes synchronized multi-modal signals such as 4D radar, in-cabin steering wheel status, brake signals, and driver-facing cameras. We hope this work will benefit the broader community working on aerial-ground collaborative perception. Future work includes refining the UAV LiDAR dataset, collecting larger-scale datasets with dense multi-UAV point cloud collaboration scenarios, expanding data collection under various weather conditions, and annotating for additional vision-action tasks. However, misuse of the dataset or models trained on it could lead to overreliance on imperfect perception systems in safety-critical applications, especially given the inclusion of time delays and pose errors that reflect real-world challenges. This could potentially result in incorrect decisions in autonomous driving scenarios. We emphasize the importance of responsible use, adherence to ethical guidelines, and the implementation of safeguards to mitigate such risks.

## Acknowledgments and Disclosure of Funding

This work was supported by the National Science and Technology Major Project (2022ZD0117901), the National Natural Science Foundation of China (62206015, 62376024), the Beijing Natural Science Foundation (L257003), and the Young Scientist Program of the National New Energy Vehicle Technology Innovation Center (Xiamen Branch). We thank the anonymous reviewers for insightful discussions.

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

# A Technical Appendices and Supplementary Material

## A.1 Coordinate Systems and Transformation

To achieve spatial synchronization between different sensors, vehicle-vehicle-UAV collaboration requires using sensor parameter information to perform coordinate system transformations. The relationships between the coordinate systems are illustrated in Fig. S 6.

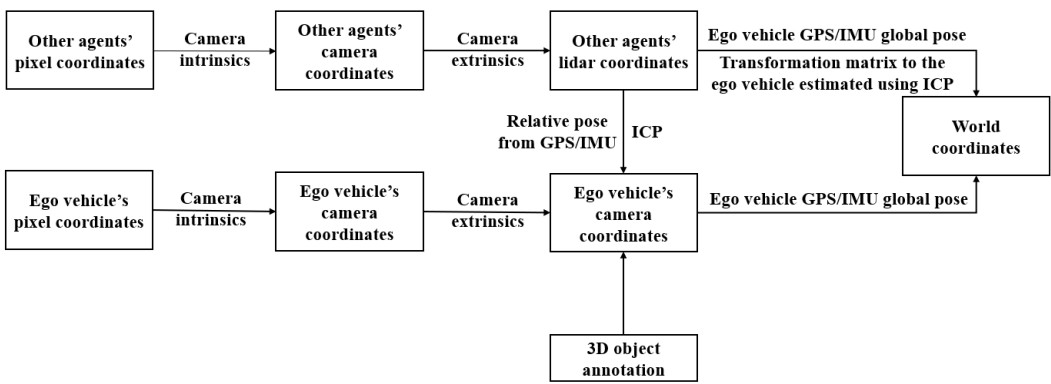

Figure 6: Relationship between coordinate systems.

**Pixel Coordinates.** The pixel coordinate system refers to a two-dimensional coordinate system defined on the image plane, typically represented as $(u, v)$, with units in pixels. In this system, the origin is located at the top-left corner of the image, the $u$-axis points to the right along the horizontal direction, and the $v$-axis points downward along the vertical direction. This coordinate system is used to describe the position of points on the two-dimensional image captured by the camera.

A 3D point in the camera coordinate system, denoted as $(x_c, y_c, z_c)$, can be projected onto the pixel coordinate system through the camera's intrinsic matrix. The transformation process can be expressed as:

$$\begin{bmatrix} u \\ v \\ 1 \end{bmatrix} = \begin{bmatrix} f_x & 0 & c_x \\ 0 & f_y & c_y \\ 0 & 0 & 1 \end{bmatrix} \begin{bmatrix} \frac{x_c}{z_c} \\ \frac{y_c}{z_c} \\ 1 \end{bmatrix} \tag{1}$$

where $f_x$ and $f_y$ represent the focal lengths along the image's $x$ and $y$ axes (in pixel units), and $(c_x, c_y)$ denote the principal point (the intersection of the optical axis with the image plane, in pixel coordinates).

**Camera Coordinate System and LiDAR-to-Camera Calibration.** The camera coordinate system is defined as a three-dimensional right-handed Cartesian coordinate system, with its origin located at the optical center of the camera. In this system, the $x$-axis points to the right along the image plane, the $y$-axis points downward along the image plane, and the $z$-axis extends forward along the optical axis of the camera.

To determine the spatial relationship between the LiDAR and each camera, we employed a point correspondence-based calibration procedure [29, 30]. Specifically, for each individual camera view, several corresponding feature points were manually selected in both the image and the LiDAR point cloud. Based on these correspondences, an initial extrinsic transformation matrix from the camera to the LiDAR was estimated using a least-squares fitting approach.

To improve calibration accuracy, the initial matrix was further refined through iterative manual adjustment and validation by visually checking the alignment of projected LiDAR points on the image plane. In order to ensure long-term calibration reliability, considering possible sensor shifts and mechanical vibrations, this calibration procedure was performed once every four hours during continuous data collection.

The final extrinsic parameters for each camera were stored as a $4 \times 4$ homogeneous transformation matrix, representing the coordinate transformation from the LiDAR coordinate system to the corresponding camera coordinate system, as expressed by:

$$\begin{bmatrix} x_c \\ y_c \\ z_c \\ 1 \end{bmatrix} = \mathbf{T}_{\text{LiDAR2Cam}} \begin{bmatrix} x_l \\ y_l \\ z_l \\ 1 \end{bmatrix} \tag{2}$$

where $\mathbf{T}_{\text{LiDAR2Cam}}$ denotes the extrinsic matrix obtained from the calibration process, and $(x_l, y_l, z_l)$ and $(x_c, y_c, z_c)$ are the point coordinates in the LiDAR and camera coordinate systems, respectively.

A visualization of the LiDAR-to-camera calibration results for all recording platforms is provided in Fig. S7 S8 S9. The visualizations show the LiDAR point clouds projected onto the corresponding camera images using the estimated extrinsic parameters. Our dataset includes two ground vehicles, each equipped with five cameras providing full $360°$ coverage, and a UAV equipped with a single front-facing camera. The calibration results for each vehicle and the UAV are displayed separately, demonstrating the alignment quality across all viewpoints. The consistency between the projected LiDAR points and the visible object boundaries in the images effectively verifies the accuracy and robustness of our calibration process.

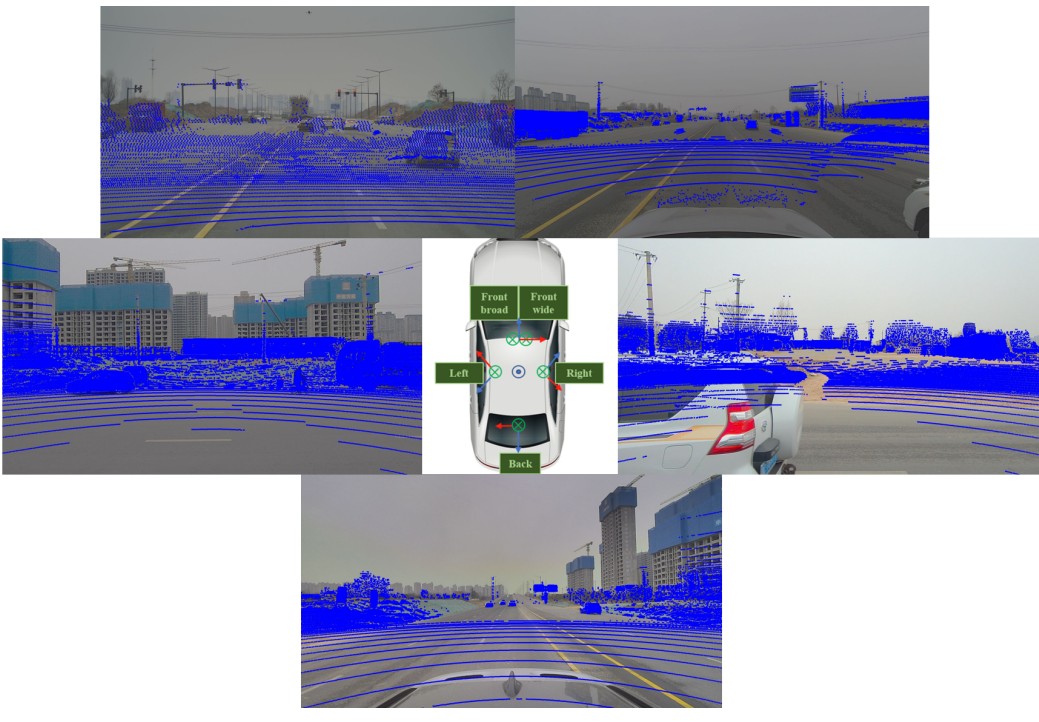

Figure 7: Visualization of the LiDAR-to-camera calibration for Ground Vehicle A equipped with five cameras covering $360°$. Projected LiDAR points align well with image features across all camera views.

**LiDAR Coordinate System and World Coordinate System.** The LiDAR coordinate system for each platform is defined relative to the sensor's installation frame on that platform. We adopt a right-handed coordinate system, where the geometric center of the LiDAR sensor is set as the origin. The x-axis points forward, the y-axis points to the left, and the z-axis points upward. The world coordinate system is established as a global East-North-Up (ENU) frame derived from GPS measurements, which provides a consistent geodetic reference for all platforms.

Point clouds collected from each platform are initially represented in their respective LiDAR coordinate systems. Using GPS and IMU data, the pose of each platform is obtained relative to the global ENU world coordinate system. In our implementation, we approximate the LiDAR-to-world

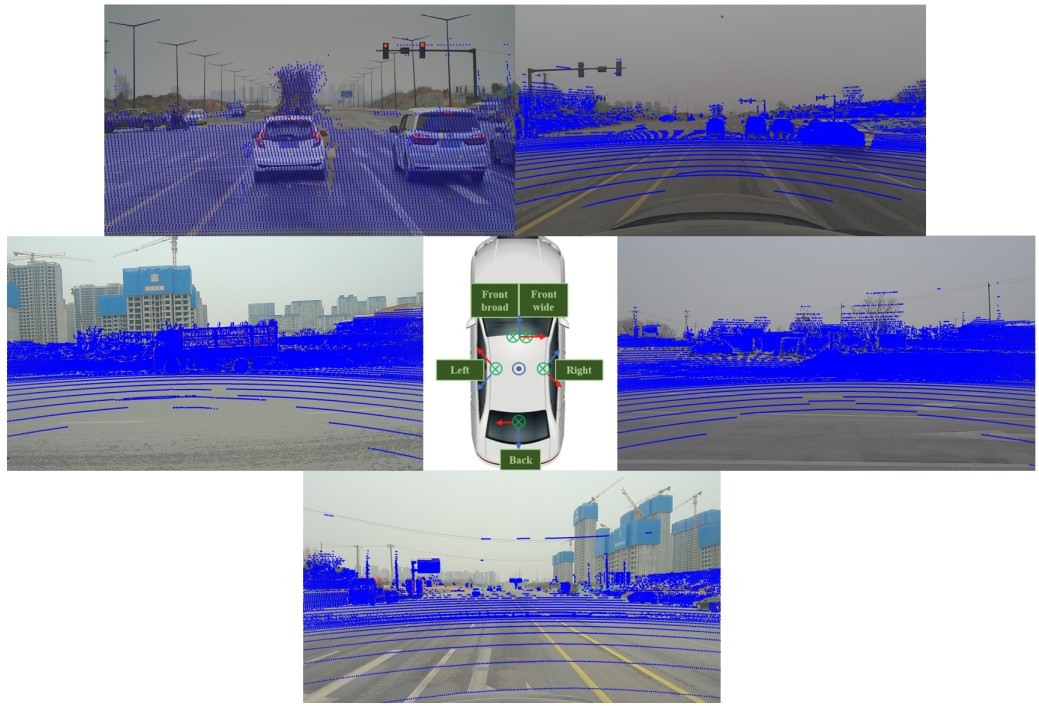

Figure 8: Visualization of the LiDAR-to-camera calibration for Ground Vehicle B equipped with five cameras covering $360°$. Projected LiDAR points align well with image features across all camera views.

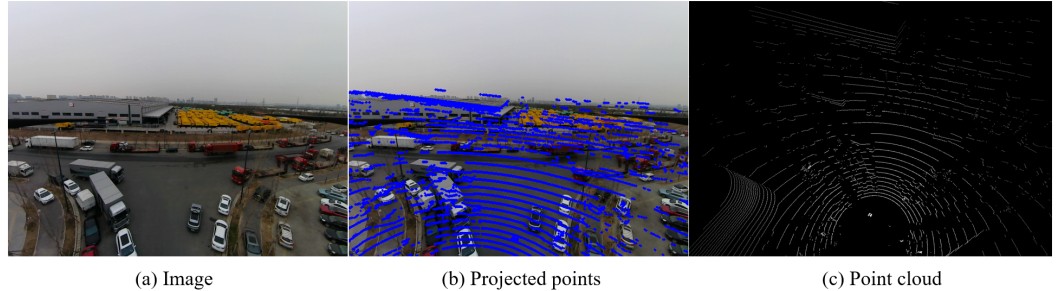

(a) Image        (b) Projected points        (c) Point cloud

Figure 9: Multi-modal data alignment from a UAV perspective. (a) Aerial image captured by the UAV-mounted camera. (b) LiDAR point cloud projected onto the image plane for visualizing alignment accuracy. (c) Top-down view of the LiDAR point cloud acquired from the UAV.

transformation using the GPS/IMU-derived vehicle pose, assuming negligible displacement between the LiDAR sensor and the localization reference point.

The transformation of a point $\mathbf{p}_{\text{lidar}}$ in the LiDAR coordinate system to the world coordinate system is performed as:

$$P_{\text{w}} \approx T_{\text{w}}^{\text{vehicle}} P_{\text{l}} \tag{3}$$

where $\mathbf{T}_{\text{w}}^{\text{vehicle}} \in SE(3)$ is the vehicle pose in the world coordinate frame obtained from GPS/IMU localization.

To compensate for residual misalignments caused by the approximation, an Iterative Closest Point (ICP) [26] algorithm is applied to refine the registration of point clouds from different platforms relative to the ego vehicle's LiDAR frame before transforming them to the world coordinate system.

The final transformation for a point cloud from another platform is given by:

$$P_{\text{w}}^{i} = T_{\text{w}}^{\text{ego}} T_{\text{ego}}^{i} P_{i} \tag{4}$$

where $\mathbf{T}_{\text{ego}}^i$ is the ICP-refined relative pose between platform $i$ and the ego vehicle.

## A.2 Multi-agent Time Synchronization

**Time Source Synchronization.** In our multi-agent system, all platforms achieve unified time source synchronization through GPS-based timing signals. Each platform's onboard clock is disciplined by the GPS receiver, providing a highly accurate and stable global time reference. This approach effectively eliminates clock drift and offset among different agents, ensuring that all sensors across vehicles and the UAV are synchronized to the same absolute time base. As a result, temporal consistency is maintained across heterogeneous sensors and platforms, which is critical for tasks such as sensor fusion, data alignment, and multi-agent cooperative perception.

**Timestamp Synchronization.** Although all platforms in our system share a common GPS-based time source, the sensors operate at different sampling frequencies, and their measurements are not necessarily captured at exactly the same timestamps. To address this, we employ the `message_filters` package in ROS to perform precise timestamp synchronization. This framework matches sensor messages based on their timestamps by finding the temporally nearest frames across heterogeneous data streams. In doing so, it compensates for both acquisition frequency differences and minor delays, ensuring accurate temporal alignment for multi-sensor fusion and multi-agent cooperative perception.

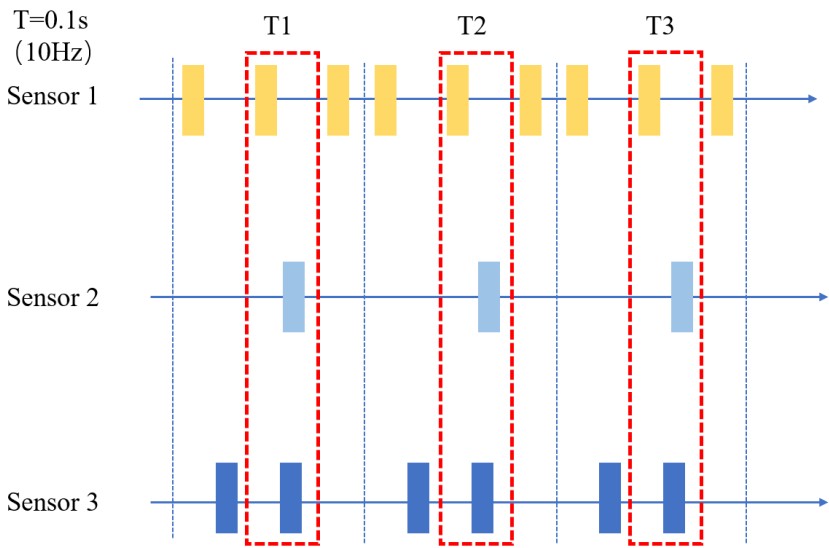

Figure 10: At 10 Hz, timestamp synchronization is performed for sensor data with different frequencies. The nearest frame within the red dashed box is regarded as the data corresponding to the same timestamp within this sampling period.

The combination of GPS-based time source synchronization and message-level timestamp synchronization enables reliable multi-sensor fusion and cooperative perception across heterogeneous platforms.

## A.3 AGC-Drive Dataset Statistics

**3D Bounding Box Category Distribution.** To provide a comprehensive overview of the dataset, we present the number of annotated 3D bounding boxes for each object category. The dataset defines a total of 13 categories, which we group into two main groups: Vehicle and Other. The Vehicle group includes four subcategories: *Car*, *Bus*, *Truck*, and *Van*, while the Other group covers nine subcategories: *Person*, *Bicycle*, *Tricycle*, *Motorcycle*, *Rider*, *Traffic Sign*, *Barrier*, *Cone*, and *Others*.

The detailed number of 3D bounding boxes for each subcategory is illustrated in Fig. S11. As shown in the figure, *Car* is the most frequently annotated category with over 650K instances, followed by *Sign*, *Truck*, and *Rider*. This distribution reflects the typical composition of cooperative driving environments, which feature a high density of vehicles and static traffic infrastructures like traffic

signs and barriers. In comparison, dynamic vulnerable road users such as *Bicycles*, *Motorcycles*, and *Persons* are less commonly observed.

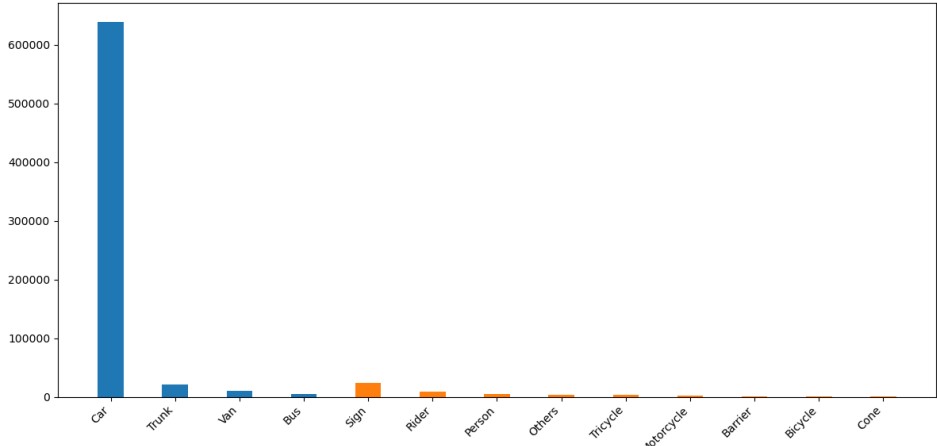

Figure 11: The number of annotated 3D bounding boxes for each object subcategory in our dataset.

## A.4   AGC-Drive vs. CoPeD

Table S5 summarizes the comparison between AGC-Drive and CoPeD [16] datasets. Both datasets provide real-world, multi-agent cooperative perception data, integrating LiDAR, camera, and GNSS/IMU sensors to support collaborative tasks in diverse environments. Additionally, both support ground and aerial agents, enabling cross-platform multi-robot cooperation.

However, significant differences exist. AGC-Drive focuses on real driving environments (rural, urban, highway) with higher vehicle speeds, while CoPeD covers mixed indoor and outdoor robot scenarios at lower speeds. AGC-Drive uniquely offers aerial LiDAR data from UAVs, in-cabin cameras, and 3D bounding box annotations with occlusion labels, providing richer multi-view and multi-modal data. In contrast, CoPeD provides 2D bounding boxes only and relies on automatic annotation methods. Furthermore, AGC-Drive contributes a larger scale of point clouds and images, with available source code, enhancing its value as an open benchmark for autonomous driving research.

Table 5: Comparison between CoPeD and AGC-Drive.

|  | AGC-Drive | CoPeD |
| --- | --- | --- |
| Source | Real | Real |
| scenario types | 14 Diverse driving scenarios | Mixed indoor and outdoor environments |
| Agents | 2*Veh & 1*UAV | 3*Ground robots & 2*Aerial robots |
| Sensors | Camera, Lidar, IMU/GPS, Radar, In-cabin camera | Camera, Lidar, IMU/GPS |
| Aerial LiDAR Support | ✓ | × |
| Cams (/Agent) | Multiple | Single |
| Height | 15 to 20m | 2m, 2 to 10m |
| Vehicle speed | 30(Rural), 30 to 50(Urban), 80(highway) km/h | 1.8(Indoor), 5.4(Outdoor) km/h |
| Categories | 13 | - |
| Labels | 3D Boxes & Occlusion | 2D Boxes |
| Images | 360,000 | 203,400 |
| Pointclouds | 80,000 | - |
| Source code | ✓ | only calibration |

