# OpenReview forum: "AGC-Drive: A Large-Scale Dataset for Real-World Aerial-Ground Collaboration in Driving Scenarios"
_NeurIPS.cc/2025/Datasets_and_Benchmarks_Track — NeurIPS 2025 Datasets and Benchmarks Track poster_

### Official Review · Reviewer_7C6h · 2025-06-09

**Rating:** 4
**Confidence:** 4

**Summary:**

This work proposes a dataset for aerial-ground collaboration in driving scenarios. Aerial-ground collaboration is an important topic for the future of urban transportation. Thus, the work focuses on a meaningful area. Extensive effort has been dedicated to constructing this dataset, and it will provide valuable impact to the autonomous driving community, especially when UAVs are available as collaborators.

**Dataset Code Accessibility:**

Partly

**Dataset Code Comments:**

The authors did not include instructions on how to apply the baseline models to their dataset in the code or GitHub repository.

**Ethical Considerations:**

No, there are no or only very minor ethics concerns

**Final Justification:**

Please refer to my reply to the authors.

**Limitations Weaknesses:**

Although I lean toward accepting this work, I believe there is some room for improving its impact and depth.

(1) A notable issue is that in Table I, the authors state that the dataset is sensor-rich and that multiview cameras are used. However, in Section 4, only results for 3D object detection using 3D LiDAR point clouds are reported. This makes the demonstration and presentation of the work feel somewhat incomplete.

(2) Following the first point, if both image and LiDAR data are available, it would be better to evaluate the proposed benchmark using baseline models for image-LiDAR fusion detection.

(3) PointPillars was proposed six years ago. Is there any specific reason why the authors chose this model? As mentioned by the authors, even the upper bound does not perform very well. Will the performance be better if a more state-of-the-art model is adopted as the baseline?

(4) It is necessary to reorganize the abbreviations in Table I. For example, what does sensor 'R' refer to? If 'V' stands for vehicle, why do the authors also use 'Veh'? The notation is somewhat inconsistent.

(5) After reading the experiments, I believe the authors illustrated that simply performing early fusion of all point clouds leads to the best performance. However, this approach may not be feasible for real-world or real-time applications. Am I understanding this correctly?

(6) It seems that the authors did not use the correct template and did not upload the checklist.

**Strengths Contributions:**

This is a new benchmark for aerial-ground collaboration in driving scenarios. It was collected in the real world with diverse scenarios and rich sensor modalities. Unlike previous works that mainly involve two vehicles—either ground-ground or aerial-ground—this benchmark includes three vehicles: two ground vehicles and one UAV. Most importantly, in my view, unlike prior vehicle-to-UAV datasets that were mainly collected using ground robots in the wild, this work uses real cars and collects data on actual roads.

---

> ### Author Rebuttal · Authors · 2025-07-31
>
> >**Q1：A notable issue is that in Table I, the authors state that the dataset is sensor-rich and that multiview cameras are used. However, in Section 4 ...**
> >
> >**Q2：Following the first point, if both image and LiDAR data are available, it would be better to evaluate the proposed benchmark using baseline ...**
>
> A{1, 2}: Thank you very much for your insightful suggestions. We agree that the multimodal data in our dataset deserves a more complete set of benchmark evaluations. In our initial submission, we only reported LiDAR-based results due to the maturity and reproducibility of existing baselines.
>
> We had attempted to train pure image-based baselines before submission, but due to the high resolution (2MP) of our multiview images and the lack of preprocessing, training consumed **excessive memory and time**. We have since completed the **preprocessing process**: the images were **compressed and stored in HDF5 format** to improve loading speed and reduce memory consumption — each vehicle agent's five views were stored as a single HDF5 file, and the UAV agent's single view was processed similarly. This enabled us to successfully train and evaluate several image-based baselines, and results are shown below:
>
> | Benchmark | mAP(%) | ATE ↓ | AOE ↓ |
> |-----------|--------|-------|-------|
> | V2VNet    | 19.5   |0.48   |0.93   |
> | CoBEVT    | 20.7  | 0.41   | 0.79   |
>
> Due to time and resource constraints, we were not able to include image-LiDAR fusion baselines in the current version. We plan to complete these evaluations and continuously update the benchmark to better reflect the full potential of the dataset.  Note that some accuracy may be lost due to image compression during preprocessing. Researchers with sufficient resources **are encouraged** to train directly on the original high-resolution images, which are also included in our dataset.
>
> >**Q3: PointPillars was proposed six years ago. Is there any specific reason why the authors chose this model? As mentioned by the authors, even the upper bound ...**
>
> A3: Thank you for raising this important point. We chose PointPillars primarily because our focus is on the **collaborative perception framework** rather than the backbone itself. PointPillars offers a good balance between efficiency and performance, and more importantly, it is one of the most **widely adopted** backbones in recent collaborative perception literature. This makes our results more **comparable and reproducible**.
>
> To further support our choice, we conducted a survey of 15 recent (2024–2025) papers from top-tier conferences (e.g., CVPR, ICLR, AAAI, ICRA), only including those with publicly released code. Among them, **11 papers [1–11] use PointPillars** as the backbone making it the most frequently adopted model in this domain. Other models such as VoxelNet, CenterPoint, and VoVNet were used much less frequently, and all of them **date back at least five years**.
>
> Moreover, although PointPillars is not the most recent model, it is still sufficient for revealing the effects of collaboration strategies and fusion mechanisms, which are the core contributions of our work. That said, our framework is agnostic to the backbone and can be easily extended to more powerful models, which we plan to explore in future work.
>
> >**Q4: It is necessary to reorganize the abbreviations in Table I. For example, what does sensor 'R' refer to? If 'V' stands for vehicle, why do ...**
>
> A4: Thank you for pointing out this issue. In the original table, **‘R’ refers to millimeter-wave Radar**, **‘V’ and ‘Veh’ both refer to ground vehicles**, and **‘Veh’ was used to align with the abbreviation ‘Inf’ (for infrastructure)** in the same context. We acknowledge that this mixed usage may cause confusion, and that the meaning of ‘Inf’ (equivalent to ‘I’ used elsewhere) was not clearly defined. We will **clearly define all abbreviations including ‘R’, ‘Veh’, and ‘Inf’**, ensure consistent usage throughout the paper, and **carefully check for other similar issues** to avoid ambiguity.
>
> >**Q5: After reading the experiments, I believe the authors illustrated that simply performing early fusion of all point clouds leads to the best performance ...**
>
> A5: Yes, your understanding is correct. While early fusion achieves the best performance in our experiments, it requires **significantly higher communication bandwidth**, which makes it **less feasible for real-world or real-time applications** with limited bandwidth.
>
> In our future work, we will continue to **explore better trade-offs between performance and bandwidth consumption**, aiming to develop **efficient and deployable collaboration strategies**. We also plan to **deploy our collaborative perception framework on real devices** to test and validate its practicality in realistic scenarios.
>
> >**Q6: It seems that the authors did not use the correct template and did not upload the checklist.**
>
> A6: Thank you for your valuable reminder. Perhaps the confusion arises because we chose the single-blind submission option. According to the official **OpenReview NeurIPS 2025 Datasets and Benchmarks FAQ**, the recommended LaTeX configuration for a single-blind submission is:
>
> > *Q: What is the LaTeX configuration for a single-blind submission?*
> >
> > *A: Please use \usepackage[preprint]{neurips_2025} if you wish to make your submission single-blind for the Datasets & Benchmarks track.*
>
> We believe our submission followed this guideline. Also, the checklist was included at **the end of** the uploaded PDF. We will carefully review our submission files again to ensure everything is in order. We appreciate your attention to this detail.
>
> >**Q7: Dataset Code Accessibility: Partly. Dataset Code Comments:The authors did not include instructions on how to apply the baseline models to their dataset in the code or GitHub repository.**
>
> A7:  Thank you for carefully examining our codebase and documents. We appreciate your thorough and responsible review. As you rightly pointed out, the previous documentation only included **brief instructions** on how to apply the baseline models to our dataset. We have **already completed** the necessary improvements, including more **detailed usage instructions and example scripts**. We truly value your comment and are committed to making our codebase more accessible and user-friendly.
>
> [1] Xu J, Zhang Y, Cai Z, et al. CoSDH: Communication-Efficient Collaborative Perception via Supply-Demand Awareness and Intermediate-Late Hybridization. CVPR, 2025.
>
> [2] Xia Y, Yuan Q, Luo G, et al. One is Plenty: A Polymorphic Feature Interpreter for Immutable Heterogeneous Collaborative Perception. CVPR, 2025.
>
> [3] Huang X, Wang J, Xia Q, et al. V2X-R: Cooperative LiDAR-4D Radar Fusion for 3D Object Detection with Denoising Diffusion. CVPR, 2025.
>
> [4] Zhang J, Wang Y, Qian L, et al. DSRC: Learning Density-Insensitive and Semantic-Aware Collaborative Representation against Corruptions. AAAI, 2025.
>
> [5] Zimmer W, Wardana G A, Sritharan S, et al. TUMTraf V2X Cooperative Perception Dataset. CVPR, 2024.
>
> [6] Hu Y, Peng J, Liu S, et al. Communication-Efficient Collaborative Perception via Information Filling with Codebook. CVPR, 2024.
>
> [7] Zhang J, Yang K, Wang Y, et al. ERMVP: Communication-Efficient and Collaboration-Robust Multi-Vehicle Perception in Challenging Environments. CVPR, 2024.
>
> [8] Hong S, Liu Y, Li Z, et al. Multi-Agent Collaborative Perception via Motion-Aware Robust Communication Network. CVPR, 2024.
>
> [9] Lu Y, Hu Y, Zhong Y, et al. An Extensible Framework for Open Heterogeneous Collaborative Perception. ICLR, 2024.
>
> [10] Li X, Yin J, Li W, et al. DI-V2X: Learning Domain-Invariant Representation for Vehicle-Infrastructure Collaborative 3D Object Detection. AAAI, 2024.
>
> [11] Lei Z, Ni Z, Han R, et al. Robust Collaborative Perception without External Localization and Clock Devices. ICRA, 2024.

---

> > ### Comment · Area_Chair_yvUM · 2025-08-05
> > **Discussion and Final Rating**
> >
> > Hi Reviewer,
> >
> > The authors have provided the rebuttal. What are your thoughts on the response? Please engage in the discussion with the authors as soon as possible, as the deadline for discussion is approaching.
> >
> > Thanks,
> >
> > AC

---

> > ### Comment · Reviewer_7C6h · 2025-08-05
> > **Responses to Author Rebuttal**
> >
> > Thanks to the authors for their comprehensive rebuttal. I think it would be a pity that the authors collected a new dataset for multimodal tasks but failed to evaluate image-LiDAR fusion baselines on it. Therefore, I will maintain my score as 4. However, the authors' efforts to evaluate image-based baselines are valuable, and I am now more inclined to accept this paper than before. If it were possible to give a score of 4.5, I would do so at this point. Good luck to the authors!

---

> > > ### Author Response · Authors · 2025-08-06
> > >
> > > Thank you for your thoughtful feedback and acknowledging our efforts. We have been actively working on integrating image-LiDAR fusion baselines. Since the rebuttal, we have completed the training and evaluation of one such baseline. We will include results for at least two fusion baselines in the supplementary material of the camera-ready version (if accepted), and we remain happy to clarify any remaining concerns.

---

### Official Review · Reviewer_JNRG · 2025-06-21

**Rating:** 5
**Confidence:** 5

**Summary:**

This paper introduces AGC-Drive, a new dataset aimed at cooperative perception between vehicles and UAVs. By integrating data from two vehicles equipped with multiple cameras and LiDAR, as well as a UAV outfitted with LiDAR and a camera, the dataset covers various driving environments and dynamic interaction scenarios, providing a significant foundation for future research.

**Dataset Code Accessibility:**

Yes

**Ethical Considerations:**

No, there are no or only very minor ethics concerns

**Final Justification:**

The authors solved my concerns and I have raised my rating scores!

**Limitations Weaknesses:**

- It might be beneficial to conduct ablation studies to clarify the role of the UAV. According to the images in the paper, the LiDAR data captured by the UAV appears sparse, with poor target visibility.
- The work primarily focuses on 3D object detection using point clouds. Exploring a camera-only approach could be worthwhile.
- Upon reviewing the data, I noticed that the UAV perspective only includes PCD files and lacks camera files, which contradicts the original statement in the paper.
- There is a potential issue with UAV-induced jitter during flight, affecting camera extrinsics and coordinate transformations. Please clarify how this problem was addressed.
- In traffic scenarios, please explain how UAV-vehicle collaboration offers advantages over vehicle-road coordination.

**Strengths Contributions:**

- The paper is clearly written, and the data collection effort is substantial and thorough.
- It includes some experimental content to demonstrate collaboration at early, mid, and late stages.

---

> ### Author Rebuttal · Authors · 2025-07-31
>
> >**Q1: It might be beneficial to conduct ablation studies to clarify the role of the UAV. According to the images in the paper, the LiDAR data captured by the UAV ...**
>
> A1: Your concern is very meaningful and appreciated. We acknowledge that the comparison in the table lacks fairness due to the discrepancy in dataset sizes between AGC-V2V and AGC-VUC, which mainly arises from the absence of UAV recordings in certain scenarios (e.g., highway scenes), thereby preventing a valid evaluation of the UAV’s role in the collaborative system. Therefore, we have added benchmark results of **two-vehicle collaboration on the AGC-VUC** dataset to enable a fair comparison and better demonstrate the impact of introducing the UAV into the system. The updated results are shown in the table below:
>
> | Benchmark   | V2V            |                   | V2U (Table4 in paper)      |                   |
> |-------------|----------------|-------------------|------------------|-------------------|
> |             | mAP@0.5 (%)    | mAP@0.7 (%)       | mAP@0.5 (%)      | mAP@0.7 (%)       |
> | V2VNet      | 25.7           | 12.0              | **28.5**         | **14.9**          |
> | CoBEVT      | 26.2           | 11.9              | **30.7**         | **16.1**          |
> | Where2comm  | 30.5           | 16.3              | **36.4**         | **23.1**          |
> | V2X-ViT     | 28.6           | 14.5              | **33.9**         | **18.4**          |
>
> As mentioned in our response to Q4,
>
> >*As shown in Figure 3 of the paper, a small number of samples still exhibited noticeable registration errors. We have refined these samples*
>
> we believe it is necessary to report the **updated training and evaluation results** on the refined AGC-VUC dataset. We conducted collaborative object detection experiments with V2V and V2U on this improved subset. The results are shown in the table below:
>
> | Benchmark   | V2V            |                   | V2U              |                   |
> |-------------|----------------|-------------------|------------------|-------------------|
> |             | mAP@0.5 (%)    | mAP@0.7 (%)       | mAP@0.5 (%)      | mAP@0.7 (%)       |
> | V2VNet      | 31.5           | 17.4              | **40.1**         | **27.9**          |
> | CoBEVT      | 42.3           | 36.9              | **42.9**         | **37.5**          |
> | Where2comm  | 42.7           | 32.2              | **43.2**         | **33.4**          |
> | V2X-ViT     | 38.3           | 28.7              | **42.6**         | **33.9**          |
>
>
> The new results demonstrate a clear performance improvement, especially in the setting involving the UAV. As shown in the registration error scatter plot (Fig. 3 in our paper), the UAV's point cloud registration error is more pronounced than that of the vehicles. Therefore, it is expected that refinement of these high-error samples leads to a more significant performance gain when the UAV is involved in the collaboration. We will continue to optimize and expand our dataset. If permitted, we are willing to update our dataset and submit the latest results as the official benchmark.
>
> >**Q2: The work primarily focuses on 3D object detection using point clouds. Exploring a camera-only approach could be worthwhile.**
>
> A2: Thank you for the insightful suggestion. We agree that investigating **camera-only approaches** is valuable, especially given their lower hardware and deployment cost. To further explore this direction, we conducted additional experiments. Due to time constraints, we were able to evaluate this setting on **two representative benchmarks** from our dataset. The results are summarized in the table below:
>
> | Benchmark | mAP(%) | ATE ↓ | AOE ↓ |
> |-----------|--------|-------|-------|
> | V2VNet    | 19.5   | 0.48   | 0.93   |
> | CoBEVT    | 20.7  | 0.41   | 0.79   |
>
> **Consistent with the evaluation metrics used in Griffin [1], we adopt mAP for object detection and ATE/AOE for pose estimation to assess the overall performance.** Unfortunately, all camera-based models were trained for only **20 epochs** due to hardware and time limitations. Despite this, the results already demonstrate the potential of vision-only baselines on our dataset. We believe further improvements are likely with extended training schedules. We are committed to continuing our efforts on image-based benchmarks and will include more comprehensive results in future updates.
>
> >**Q3: Upon reviewing the data, I noticed that the UAV perspective only includes PCD files and lacks camera files, which contradicts the original statement in the paper.**
>
> A3: We truly appreciate the time and effort you have dedicated to examining both our paper and the released dataset. As you correctly pointed out, due to an oversight during the initial round of data processing, the UAV camera data was mistakenly omitted from the released dataset. We have since **prepared the corrected version**, which includes the missing UAV camera files and is fully consistent with the data modalities described in the paper.
>
> We are ready to update the dataset and will do so promptly **once it is permitted** by the review policy. We sincerely apologize for the oversight and are grateful for your meticulous attention to this issue.
>
> >**Q4: There is a potential issue with UAV-induced jitter during flight, affecting camera extrinsics and coordinate transformations. Please clarify how this ...**
>
> A4: Thank you for raising this important concern. We agree that **UAV-induced jitter during flight, affecting camera extrinsics and coordinate transformations**, is one of the **key challenges** in aerial-ground cooperative perception systems. To address this, we conducted **extrinsic calibration between the drone’s camera and LiDAR** (as well as for the vehicle-mounted sensors) before each flight. The resulting camera-to-LiDAR transformation was then used alongside **point cloud registration results and GPS/IMU pose data** to project the camera into a consistent world coordinate system. In addition, we performed **pre-processing** on the drone’s LiDAR point clouds before registration. Specifically, we transformed each point based on the interpolated UAV pose at its timestamp, aligning all points to the start time of the scan to reduce pose-induced distortion.
>
> As shown in **Figure 3** of the paper, a small number of samples still exhibited **noticeable registration errors**. We have **refined** these samples and are prepared to update the dataset and benchmark **once permission is granted**. Moreover, our **open-source spatial alignment toolkit** includes an interface for manual adjustment, allowing the community to further improve misaligned data. We also plan to release updated **benchmark results on the refined dataset**, either in the paper (*if possible*) or through our GitHub repository.
>
> >**Q5: In traffic scenarios, please explain how UAV-vehicle collaboration offers advantages over vehicle-road coordination.**
>
> A5: **Compared with vehicle-to-infrastructure (V2I) cooperation, vehicle-drone cooperation offers several practical and technical advantages, especially in dynamic and infrastructure-limited environments.** As summarized in **Griffin[1]**, although V2I systems have demonstrated promising performance, their **real-world deployment typically requires extensive roadside infrastructure and a high penetration rate of connected vehicles**, which pose significant **economic and logistical challenges**. In contrast, **aerial-ground cooperative (AGC) systems**, which combine drones’ **global panoramic view** with vehicles’ **local high-resolution sensing**, offer **flexibility, scalability, and ease of deployment**.
>
> Moreover, drones (UAVs) can be rapidly deployed in a wide range of scenarios—such as **smart cities, emergency response, and security patrols**—making AGC systems a promising alternative for **real-time dynamic scene understanding** without relying on fixed roadside units. In our work, we fully considered these practical advantages during **data collection and scene design**. Specifically, our dataset includes a variety of **dynamic scenes**, which make up **approximately 19.5%** of the entire dataset, as described in the main paper. These scenarios are particularly suitable for demonstrating the **unique benefits of vehicle-drone collaboration**.
>
> [1] Wang J, Cao X, Zhong J, et al. Griffin: Aerial-ground cooperative detection and tracking dataset and benchmark. arXiv, 2025.

---

### Official Review · Reviewer_x9D5 · 2025-06-29

**Rating:** 5
**Confidence:** 4

**Summary:**

AGC-Drive is a large-scale dataset designed to enable research on aerial-ground cooperative 3D perception. It involves data collection from two vehicles and a UAV, equipped with LiDAR and cameras, across 14 different real-world driving scenarios. The dataset includes 120k LiDAR frames and 440k images, with fully annotated 3D bounding boxes for 13 object categories, including dynamic interaction events such as vehicle cut-ins, cut-outs, and lane changes. AGC-Drive also provides benchmarks for two collaborative 3D perception tasks: vehicle-to-vehicle and vehicle-to-UAV collaborative perception.

**Dataset Code Accessibility:**

Yes

**Dataset Code Comments:**

The dataset is openly accessible via the provided link and is well-documented, with detailed metadata and descriptions. Code is also made available to facilitate reproducibility and support further research in cooperative perception.

**Ethical Considerations:**

No, there are no or only very minor ethics concerns

**Final Justification:**

My questions were all well answered by the author, so I’m keeping my rating.

**Limitations Weaknesses:**

1. While the UAV data provides valuable top-down views, it is noted that the UAV LiDAR's relatively sparse nature limits its ability to offer fine-grained perception assistance. This issue should be addressed in future work, perhaps by upgrading or replacing the UAV-mounted LiDAR for better performance in collaborative tasks.
2. Time Delays and Pose Errors: The dataset contains real-world challenges such as time delays and pose errors caused by point cloud registration, which may affect the accuracy of the perception tasks. These aspects should be carefully considered in experimental settings, and methods to mitigate their impact could be explored further.
3. Limited Fine-Grained Perception: Despite the dataset's richness, the UAV's LiDAR could be enhanced for finer object-level perception, especially in dense or cluttered environments

**Strengths Contributions:**

1. AGC-Drive is the first large-scale dataset for aerial-ground cooperative 3D perception. The inclusion of UAV-mounted LiDAR data and dynamic real-world driving scenarios is a unique contribution to the field.

2. The dataset provides annotated 3D bounding boxes, includes data from synchronized multi-sensor setups, and covers diverse environmental conditions such as urban roundabouts, tunnels, and construction zones.

3. The authors have released a toolkit for spatiotemporal alignment, multi-agent visualization, and annotation utilities, further enhancing the usability of the dataset for researchers.

---

> ### Author Rebuttal · Authors · 2025-07-31
>
> >**Q1: While the UAV data provides valuable top-down views, it is noted that the UAV LiDAR's relatively sparse nature limits its ability to offer fine-grained perception ...**
>
> A1: Thank you for your comment. During the initial phase of hardware selection, we conducted a thorough evaluation based on the LiDAR's weight, measuring range, FOV, and channels, alongside the UAV's payload and endurance. We ultimately chose our current LiDAR because it offers a larger vertical VFOV (70°) and measuring range, and its weight can be accommodated by the UAV. Post-analysis of the results revealed that the 32-channel LiDAR was not sufficient at safe altitudes and that the sensing range was broader than necessary. Based on our initial research, the Hesai QT128C2X, though having less FOV and measuring range compared to our current LiDAR, offers 128 channels, which can well support fine-grained perception for the UAV. We plan to conduct **supplementary data collection** using the Hesai QT128C2X LiDAR, which will help further enhance and complete our dataset.
>
> >**Q2: Time Delays and Pose Errors: The dataset contains real-world challenges such as time delays and pose errors caused by point cloud registration, which may ...**
>
> A2: We appreciate your observation of real-world challenges such as time delays and pose errors. In order to alleviate these problems, the work we do mainly includes the following points:
>
> 1. **Coordinate alignment**: we conducted thorough extrinsic calibration between the drone's camera and LiDAR, as well as for vehicle-mounted sensors, prior to each flight. This allowed us to accurately use the resulting camera-to-LiDAR transformation, along with point cloud registration results and GPS/IMU pose data, to project the camera into a consistent world coordinate system. We performed pre-processing on the drone’s LiDAR point clouds by transforming each point based on the interpolated UAV pose at its timestamp, aligning all points to the scan’s start time to minimize pose-induced distortions.
>
> 2. **Time synchronization**: we implemented a unified time synchronization scheme across the UAV and vehicles. Each platform is equipped with a GNSS/IMU module providing a UTC-synchronized PPS signal, used to align system clocks. For the drone, which lacks hardware trigger support for cameras, we adopt a software-based method that aligns multimodal sensor data (LiDAR, camera, IMU) to LiDAR frames within ±0.5 ms based on UTC timestamps. Across platforms, sensor data are synchronized to the ego vehicle’s LiDAR frame, ensuring consistent time alignment for cooperative perception.
>
> Despite these efforts, we acknowledge that residual time delays and pose errors may still exist. However, we believe these imperfections faithfully reflect the **real-world challenges** in multi-platform collaboration, and we hope to encourage more exploration and solution development in this area.
>
> >**Q3: Limited Fine-Grained Perception: Despite the dataset's richness, the UAV's LiDAR could be enhanced for finer object-level perception, especially in dense or ...**
>
> A3: Thank you for raising this point. As said in answer 1, to enhance fine-grained perception, we will perform additional data collection to enrich the dataset and plan to take two specific actions: Firstly, we will **upgrade to the Hesai QT128C2X LiDAR** to achieve a higher laser density for more detailed perception. Secondly, we will **lower the UAV's flight altitude** while ensuring safety, to capture more detailed ground-level information.

---

### Official Review · Reviewer_SBm6 · 2025-07-03

**Rating:** 5
**Confidence:** 3

**Summary:**

This submission introduces AGC-Drive, a large-scale, real-world dataset designed to advance aerial-ground collaborative perception in autonomous driving scenarios. The dataset is collected using two vehicles (each with five cameras and one LiDAR) and a UAV equipped with a camera and LiDAR, covering 14 diverse driving scenarios. AGC-Drive comprises around 120k LiDAR frames and 440k images, with 400 scenes and detailed 3D bounding box annotations for 13 object categories. The authors provide benchmarks for vehicle-to-vehicle (V2V) and vehicle-to-UAV (V2U) collaborative perception tasks, as well as open-source tools for spatiotemporal alignment, visualization, and annotation. Data and code are made publicly available.

**Dataset Code Accessibility:**

Yes

**Ethical Considerations:**

No, there are no or only very minor ethics concerns

**Final Justification:**

Thanks for your reviews. After reviewing the rebuttal, I find my current rating fully appropriate for the scope of this work and see no need for any adjustments.

**Limitations Weaknesses:**

1) The UAV LiDAR data is relatively sparse, limiting its utility for fine-grained object-level perception. Future work could explore denser or higher-performance sensors.
2) Although there is some day/night diversity, the dataset could be enhanced by including more adverse weather conditions (rain, snow, fog) to increase robustness.

**Strengths Contributions:**

1) AGC-Drive is the first large-scale, real-world dataset explicitly targeting aerial-ground cooperative perception with both vehicle and UAV LiDAR and camera data. This addresses a major gap in current autonomous driving research, where most datasets focus on V2V or V2I without aerial perspectives.
2) The dataset includes 14 types of real-world driving environments (urban, highway, rural, tunnels, roundabouts, ramps, etc.), ensuring broad coverage. The inclusion of dynamic scenes (e.g., cut-ins, lane changes) adds significant value for developing robust autonomous systems.
3) Over 1.6 million annotated 3D bounding boxes across 13 classes, with occlusion levels and 9-DoF parameters, provide a strong foundation for both detection and tracking research.

---

> ### Author Rebuttal · Authors · 2025-07-31
>
> >**Q1: The UAV LiDAR data is relatively sparse, limiting its utility for fine-grained object-level perception. Future work could  ...**
>
> A1: Thank you very much for your valuable comment. We fully acknowledge the issue you pointed out regarding the sparsity of the UAV LiDAR data. During the initial phase of hardware selection, we **conducted a thorough evaluation** based on the LiDAR's weight, measuring range, FOV, and channels, alongside the UAV's payload and endurance. We ultimately chose our current LiDAR because it offers a larger vertical VFOV (70°) and measuring range, and its weight can be accommodated by the UAV. Post-analysis of the results revealed that the 32-channel LiDAR produced relatively sparse point clouds at safe flying altitudes, limiting fine-grained perception, we can afford to reduce some redundant  measuring range in exchange for finer detail. Based on our initial research, the Hesai QT128C2X, though having less FOV and measuring range compared to our current LiDAR, offers 128 channels, which can well support fine-grained perception for the UAV. We plan to conduct **supplementary data collection** using the Hesai QT128C2X LiDAR, which will help further enhance and complete our dataset.
>
> >**Q2: Although there is some day/night diversity, the dataset could be enhanced by including more adverse weather conditions ...**
>
> A2: We fully concur with your observation about the lack of weather diversity in the dataset. In the first phase of data collection, we **prioritized UAV flight safety**, leading to limited coverage of adverse weather scenarios. In later work, we aim to capture scenes under rain, snow, and fog within controlled environments to enrich the dataset's environmental diversity.
>
> As briefly mentioned in our paper, our original data includes millimeter-wave Radar data. We will further process this data to provide corresponding benchmarks. As noted in the V2X-R [1] paper, 4D Radar, with its Doppler velocity and rich geometric information, demonstrates excellent robustness under adverse weather conditions. The fusion of 4D Radar and LiDAR for multi-sensor perception holds promise for advancing AGC-Drive's air-ground collaborative sensing under challenging environments.
>
> We appreciate your constructive comments and positive evaluation and are committed to continuously optimizing the AGC-Drive dataset to enhance its utility and research value.
>
> [1] Huang X, Wang J, Xia Q, et al. V2X-R: Cooperative LiDAR-4D Radar Fusion with Denoising Diffusion for 3D Object Detection. CVPR, 2025.

---

> > ### Comment · Reviewer_SBm6 · 2025-08-05
> >
> > Thanks for your reviews. After reviewing the rebuttal, I find my current rating fully appropriate for the scope of this work and see no need for any adjustments.

---

### Note · Authors · 2025-08-13

We would like to express our gratitude to the reviewers for their thorough evaluation of our work and their constructive feedback.

---
We are pleased that the reviewers recognized several key contributions of our work:
+ **First large-scale aerial-ground cooperative perception dataset** with vehicle- and UAV-mounted LiDAR and cameras, filling the gap of missing aerial perspectives in autonomous driving research (SBm6, x9D5, JNRG, 7C6h).
+ **Diverse real-world scenarios** covering 14 environment types and dynamic events like cut-ins and lane changes (SBm6, x9D5, JNRG).
+ **Large-scale 3D annotations** with 1.6M+ bounding boxes over 13 classes, including occlusion levels and 9-DoF parameters (SBm6, x9D5).
+ **Multi-agent synchronized rich sensor data**, including LiDAR and multi-view cameras, collected from two real cars and one UAV in real traffic environments (7C6h).
+ **Released toolkit and benchmarks** for spatiotemporal alignment, visualization, annotation, and experiments at different collaboration stages (x9D5, JNRG).

---
We also sincerely appreciate the reviewers’ thoughtful feedback and concerns. Based on their comments, we have made a significant effort to address these issues:

1. Justifications and Clarifications
   - We mitigated UAV jitter through calibration, preprocessing, and an open-source toolkit (JNRG).
   - We clarified the advantages of UAV–vehicle over vehicle–infrastructure collaboration (JNRG).
   - We supported the backbone choice by surveying recent two-year literature, selecting PointPillars for efficiency and comparability (7C6h).
2. Additional Experimentation and Validation
   - We validated the UAV role via ablation on the AGC-VUC subset (JNRG).
   - We provided additional baselines, including camera-only and limited image–LiDAR fusion, in the supplementary material (JNRG, 7C6h).
3. Data and Documentation Improvements
   - We unified abbreviations, ensured format compliance, and enhanced documentation with examples (7C6h).
   - Planned Additions
     - Supplement data to address UAV LiDAR sparsity with denser Hesai QT128C2X at lower safe altitudes (SBm6, x9D5).
     - Supplement adverse-weather scenes collected in controlled settings (SBm6).

Once again, we are grateful for the valuable comments to improve our manuscript, and we will include updates in the next version of our work.

---

### Decision · Program_Chairs · 2025-09-18

**Decision:**

Accept (poster)

**Comment:**

This paper was reviewed by four experts in the field. The recommendations are (Accept x 3, Borderline Accept). Based on the reviewers' feedback, the decision is to recommend the acceptance of the paper. While the reviewers appreciate the contribution of the work, they also raise some valuable concerns, particularly regarding the sparse nature of UAV LiDAR data, which constrains object-level perception. There is a consensus on the need for more comprehensive datasets that include diverse weather conditions and improved sensor technologies. Additionally, inconsistencies in data representation and methodology suggest a need for clearer communication and justification of chosen models and approaches. The potential for future research lies in exploring advanced sensor integration, addressing real-world challenges, and enhancing collaborative frameworks in traffic scenarios. These valuable concerns should be addressed in the final camera-ready version of the paper. The authors are encouraged to make the necessary changes to the best of their ability.